# FOURIER NEURAL OPERATOR FOR PARAMETRIC PARTIAL DIFFERENTIAL EQUATIONS

**Zongyi Li**
zongyili@caltech.edu

**Nikola Kovachki**
nkovachki@caltech.edu

**Kamyar Azizzadenesheli**
kamyar@purdue.edu

**Burigede Liu**
bgl@caltech.edu

**Kaushik Bhattacharya**
bhatta@caltech.edu

**Andrew Stuart**
astuart@caltech.edu

**Anima Anandkumar**
anima@caltech.edu

## ABSTRACT

The classical development of neural networks has primarily focused on learning mappings between finite-dimensional Euclidean spaces. Recently, this has been generalized to neural operators that learn mappings between function spaces. For partial differential equations (PDEs), neural operators directly learn the mapping from any functional parametric dependence to the solution. Thus, they learn an entire family of PDEs, in contrast to classical methods which solve one instance of the equation. In this work, we formulate a new neural operator by parameterizing the integral kernel directly in Fourier space, allowing for an expressive and efficient architecture. We perform experiments on Burgers' equation, Darcy flow, and Navier-Stokes equation. The Fourier neural operator is the first ML-based method to successfully model turbulent flows with zero-shot super-resolution. It is up to three orders of magnitude faster compared to traditional PDE solvers. Additionally, it achieves superior accuracy compared to previous learning-based solvers under fixed resolution.

## 1 INTRODUCTION

Many problems in science and engineering involve solving complex partial differential equation (PDE) systems repeatedly for different values of some parameters. Examples arise in molecular dynamics, micro-mechanics, and turbulent flows. Often such systems require fine discretization in order to capture the phenomenon being modeled. As a consequence, traditional numerical solvers are slow and sometimes inefficient. For example, when designing materials such as airfoils, one needs to solve the associated inverse problem where thousands of evaluations of the forward model are needed. A fast method can make such problems feasible.

**Conventional solvers vs. Data-driven methods.** Traditional solvers such as finite element methods (FEM) and finite difference methods (FDM) solve the equation by discretizing the space. Therefore, they impose a trade-off on the resolution: coarse grids are fast but less accurate; fine grids are accurate but slow. Complex PDE systems, as described above, usually require a very fine discretization, and therefore very challenging and time-consuming for traditional solvers. On the other hand, data-driven methods can directly learn the trajectory of the family of equations from the data. As a result, the learning-based method can be orders of magnitude faster than the conventional solvers.

Machine learning methods may hold the key to revolutionizing scientific disciplines by providing fast solvers that approximate or enhance traditional ones (Raissi et al., 2019; Jiang et al., 2020; Greenfeld et al., 2019; Kochkov et al., 2021). However, classical neural networks map between finite-dimensional spaces and can therefore only learn solutions tied to a specific discretization. This is often a limitation for practical applications and therefore the development of mesh-invariant neural networks is required. We first outline two mainstream neural network-based approaches for PDEs – the finite-dimensional operators and Neural-FEM.

**Finite-dimensional operators.** These approaches parameterize the solution operator as a deep convolutional neural network between finite-dimensional Euclidean spaces Guo et al. (2016); Zhu

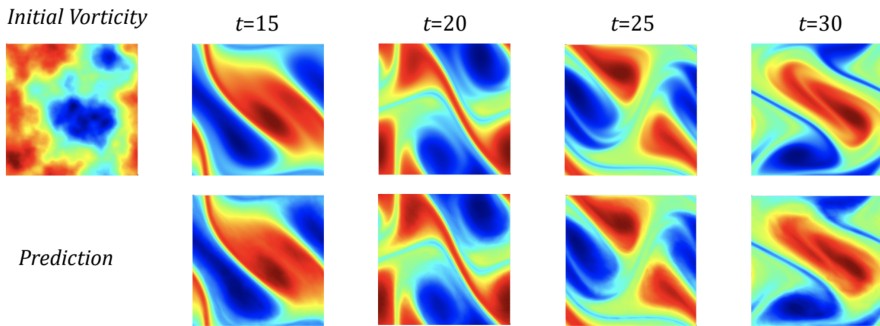

*Initial Vorticity*  t=15  t=20  t=25  t=30

*Prediction*

Zero-shot super-resolution: Navier-Stokes Equation with Reynolds number 10000; Ground truth on top and prediction on bottom; trained on $64 \times 64 \times 20$ dataset; evaluated on $256 \times 256 \times 80$ (see Section 5.4).

Figure 1: **top:** The architecture of the Fourier layer; **bottom:** Example flow from Navier-Stokes.

& Zabaras (2018); Adler & Oktem (2017); Bhatnagar et al. (2019); Khoo et al. (2017). Such approaches are, by definition, mesh-dependent and will need modifications and tuning for different resolutions and discretizations in order to achieve consistent error (if at all possible). Furthermore, these approaches are limited to the discretization size and geometry of the training data and hence, it is not possible to query solutions at new points in the domain. In contrast, we show, for our method, both invariance of the error to grid resolution, and the ability to transfer the solution between meshes.

**Neural-FEM.** The second approach directly parameterizes the solution function as a neural network (E & Yu, 2018; Raissi et al., 2019; Bar & Sochen, 2019; Smith et al., 2020; Pan & Duraisamy, 2020). This approach is designed to model one specific instance of the PDE, not the solution operator. It is mesh-independent and accurate, but for any given new instance of the functional parameter/coefficient, it requires training a new neural network. The approach closely resembles classical methods such as finite elements, replacing the linear span of a finite set of local basis functions with the space of neural networks. The Neural-FEM approach suffers from the same computational issue as classical methods: the optimization problem needs to be solved for every new instance. Furthermore, the approach is limited to a setting in which the underlying PDE is known.

**Neural Operators.** Recently, a new line of work proposed learning mesh-free, infinite-dimensional operators with neural networks (Lu et al., 2019; Bhattacharya et al., 2020; Nelsen & Stuart, 2020; Li et al., 2020b;a; Patel et al., 2021). The neural operator remedies the mesh-dependent nature of the finite-dimensional operator methods discussed above by producing a single set of network parameters that may be used with different discretizations. It has the ability to transfer solutions between meshes. Furthermore, the neural operator needs to be trained only once. Obtaining a solution for a new instance of the parameter requires only a forward pass of the network, alleviating the major computational issues incurred in Neural-FEM methods. Lastly, the neural operator requires no knowledge of the underlying PDE, only data. Thus far, neural operators have not yielded efficient numerical algorithms that can parallel the success of convolutional or recurrent neural networks in the finite-dimensional setting due to the cost of evaluating integral operators. Through the fast Fourier transform, our work alleviates this issue.

**Fourier Transform.** The Fourier transform is frequently used in spectral methods for solving differential equations, since differentiation is equivalent to multiplication in the Fourier domain. Fourier transforms have also played an important role in the development of deep learning. In theory, they appear in the proof of the universal approximation theorem (Hornik et al., 1989) and, empirically, they have been used to speed up convolutional neural networks (Mathieu et al., 2013). Neural network architectures involving the Fourier transform or the use of sinusoidal activation functions have also been proposed and studied (Bengio et al., 2007; Mingo et al., 2004; Sitzmann et al., 2020). Recently, some spectral methods for PDEs have been extended to neural networks (Fan et al., 2019a;b; Kashinath et al., 2020). We build on these works by proposing a neural operator architecture defined directly in Fourier space with quasi-linear time complexity and state-of-the-art approximation capabilities.

**Our Contributions.** We introduce the Fourier neural operator, a novel deep learning architecture able to learn mappings between infinite-dimensional spaces of functions; the integral operator is restricted to a convolution, and instantiated through a linear transformation in the Fourier domain.

- The Fourier neural operator is the first work that learns the resolution-invariant solution operator for the family of Navier-Stokes equation in the turbulent regime, where previous graph-based neural operators do not converge.

- By construction, the method shares the same learned network parameters irrespective of the discretization used on the input and output spaces. It can do zero-shot super-resolution: trained on a lower resolution directly evaluated on a higher resolution, as shown in Figure 1.

- The proposed method consistently outperforms all existing deep learning methods even when fixing the resolution to be $64 \times 64$. It achieves error rates that are $30\%$ lower on Burgers' Equation, $60\%$ lower on Darcy Flow, and $30\%$ lower on Navier Stokes (turbulent regime with Reynolds number 10000). When learning the mapping for the entire time series, the method achieves $< 1\%$ error with Reynolds number 1000 and $8\%$ error with Reynolds number 10000.

- On a $256 \times 256$ grid, the Fourier neural operator has an inference time of only $0.005s$ compared to the $2.2s$ of the pseudo-spectral method used to solve Navier-Stokes. Despite its tremendous speed advantage, the method does not suffer from accuracy degradation when used in downstream applications such as solving the Bayesian inverse problem, as shown in Figure 6.

We observed that the proposed framework can approximate complex operators raising in PDEs that are highly non-linear, with high frequency modes and slow energy decay. The power of neural operators comes from combining linear, global integral operators (via the Fourier transform) and non-linear, local activation functions. Similar to the way standard neural networks approximate highly non-linear functions by combining linear multiplications with non-linear activations, the proposed neural operators can approximate highly non-linear operators.

## 2 LEARNING OPERATORS

Our methodology learns a mapping between two infinite dimensional spaces from a finite collection of observed input-output pairs. Let $D \subset \mathbb{R}^d$ be a bounded, open set and $\mathcal{A} = \mathcal{A}(D; \mathbb{R}^{d_a})$ and $\mathcal{U} = \mathcal{U}(D; \mathbb{R}^{d_u})$ be separable Banach spaces of function taking values in $\mathbb{R}^{d_a}$ and $\mathbb{R}^{d_u}$ respectively. Furthermore let $G^\dagger : \mathcal{A} \to \mathcal{U}$ be a (typically) non-linear map. We study maps $G^\dagger$ which arise as the solution operators of parametric PDEs – see Section 5 for examples. Suppose we have observations $\{a_j, u_j\}_{j=1}^N$ where $a_j \sim \mu$ is an i.i.d. sequence from the probability measure $\mu$ supported on $\mathcal{A}$ and $u_j = G^\dagger(a_j)$ is possibly corrupted with noise. We aim to build an approximation of $G^\dagger$ by constructing a parametric map

$$G : \mathcal{A} \times \Theta \to \mathcal{U} \qquad \text{or equivalently,} \qquad G_\theta : \mathcal{A} \to \mathcal{U}, \quad \theta \in \Theta \tag{1}$$

for some finite-dimensional parameter space $\Theta$ by choosing $\theta^\dagger \in \Theta$ so that $G(\cdot, \theta^\dagger) = G_{\theta^\dagger} \approx G^\dagger$. This is a natural framework for learning in infinite-dimensions as one could define a cost functional $C : \mathcal{U} \times \mathcal{U} \to \mathbb{R}$ and seek a minimizer of the problem

$$\min_{\theta \in \Theta} \mathbb{E}_{a \sim \mu}[C(G(a, \theta), G^\dagger(a))]$$

which directly parallels the classical finite-dimensional setting (Vapnik, 1998). Showing the existence of minimizers, in the infinite-dimensional setting, remains a challenging open problem. We will approach this problem in the test-train setting by using a data-driven empirical approximation to the cost used to determine $\theta$ and to test the accuracy of the approximation. Because we conceptualize our methodology in the infinite-dimensional setting, all finite-dimensional approximations share a common set of parameters which are consistent in infinite dimensions. A table of notation is shown in Appendix 3.

**Learning the Operator.** Approximating the operator $G^\dagger$ is a different and typically much more challenging task than finding the solution $u \in \mathcal{U}$ of a PDE for a single instance of the parameter $a \in \mathcal{A}$. Most existing methods, ranging from classical finite elements, finite differences, and finite volumes to modern machine learning approaches such as physics-informed neural networks

(a)

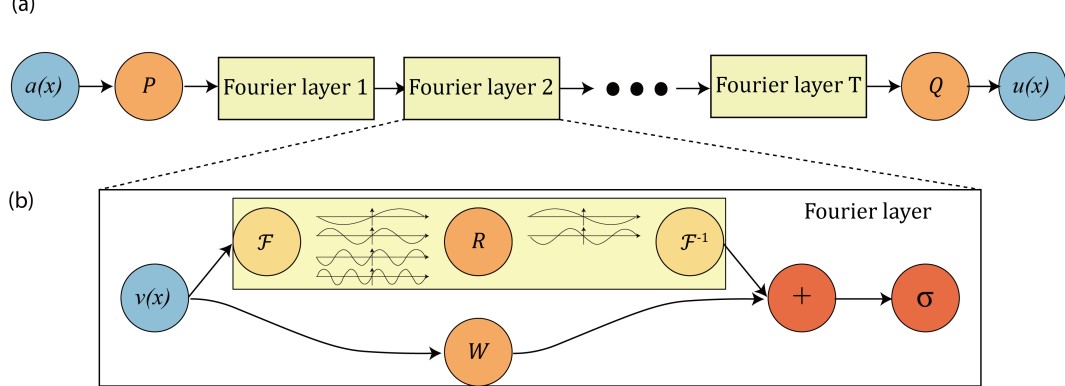

(b)

(a) **The full architecture of neural operator**: start from input $a$. 1. Lift to a higher dimension channel space by a neural network $P$. 2. Apply four layers of integral operators and activation functions. 3. Project back to the target dimension by a neural network $Q$. Output $u$. (b) **Fourier layers**: Start from input $v$. On top: apply the Fourier transform $\mathcal{F}$; a linear transform $R$ on the lower Fourier modes and filters out the higher modes; then apply the inverse Fourier transform $\mathcal{F}^{-1}$. On the bottom: apply a local linear transform $W$.

Figure 2: **top:** The architecture of the neural operators; **bottom:** Fourier layer.

(PINNs) (Raissi et al., 2019) aim at the latter and can therefore be computationally expensive. This makes them impractical for applications where a solution to the PDE is required for many different instances of the parameter. On the other hand, our approach directly approximates the operator and is therefore much cheaper and faster, offering tremendous computational savings when compared to traditional solvers. For an example application to Bayesian inverse problems, see Section 5.5.

**Discretization.** Since our data $a_j$ and $u_j$ are, in general, functions, to work with them numerically, we assume access only to point-wise evaluations. Let $D_j = \{x_1, \ldots, x_n\} \subset D$ be a $n$-point discretization of the domain $D$ and assume we have observations $a_j|_{D_j} \in \mathbb{R}^{n \times d_a}$, $u_j|_{D_j} \in \mathbb{R}^{n \times d_v}$, for a finite collection of input-output pairs indexed by $j$. To be discretization-invariant, the neural operator can produce an answer $u(x)$ for any $x \in D$, potentially $x \notin D_j$. Such a property is highly desirable as it allows a transfer of solutions between different grid geometries and discretizations.

## 3 NEURAL OPERATOR

The neural operator, proposed in (Li et al., 2020b), is formulated as an iterative architecture $v_0 \mapsto v_1 \mapsto \ldots \mapsto v_T$ where $v_j$ for $j = 0, 1, \ldots, T-1$ is a sequence of functions each taking values in $\mathbb{R}^{d_v}$. As shown in Figure 2 (a), the input $a \in \mathcal{A}$ is first lifted to a higher dimensional representation $v_0(x) = P(a(x))$ by the local transformation $P$ which is usually parameterized by a shallow fully-connected neural network. Then we apply several iterations of updates $v_t \mapsto v_{t+1}$ (defined below). The output $u(x) = Q(v_T(x))$ is the projection of $v_T$ by the local transformation $Q : \mathbb{R}^{d_v} \to \mathbb{R}^{d_u}$. In each iteration, the update $v_t \mapsto v_{t+1}$ is defined as the composition of a non-local integral operator $\mathcal{K}$ and a local, nonlinear activation function $\sigma$.

**Definition 1 (Iterative updates)** *Define the update to the representation $v_t \mapsto v_{t+1}$ by*

$$v_{t+1}(x) := \sigma\Big(W v_t(x) + \big(\mathcal{K}(a; \phi)v_t\big)(x)\Big), \qquad \forall x \in D \tag{2}$$

*where $\mathcal{K} : \mathcal{A} \times \Theta_{\mathcal{K}} \to \mathcal{L}(\mathcal{U}(D; \mathbb{R}^{d_v}), \mathcal{U}(D; \mathbb{R}^{d_v}))$ maps to bounded linear operators on $\mathcal{U}(D; \mathbb{R}^{d_v})$ and is parameterized by $\phi \in \Theta_{\mathcal{K}}$, $W : \mathbb{R}^{d_v} \to \mathbb{R}^{d_v}$ is a linear transformation, and $\sigma : \mathbb{R} \to \mathbb{R}$ is a non-linear activation function whose action is defined component-wise.*

We choose $\mathcal{K}(a; \phi)$ to be a kernel integral transformation parameterized by a neural network.

**Definition 2 (Kernel integral operator $\mathcal{K}$)** *Define the kernel integral operator mapping in (2) by*

$$\big(\mathcal{K}(a; \phi)v_t\big)(x) := \int_D \kappa\big(x, y, a(x), a(y); \phi\big)v_t(y)\mathrm{d}y, \qquad \forall x \in D \tag{3}$$

where $\kappa_\phi : \mathbb{R}^{2(d+d_a)} \to \mathbb{R}^{d_v \times d_v}$ *is a neural network parameterized by* $\phi \in \Theta_\mathcal{K}$.

Here $\kappa_\phi$ plays the role of a kernel function which we learn from data. Together definitions 1 and 2 constitute a generalization of neural networks to infinite-dimensional spaces as first proposed in Li et al. (2020b). Notice even the integral operator is linear, the neural operator can learn highly non-linear operators by composing linear integral operators with non-linear activation functions, analogous to standard neural networks.

If we remove the dependence on the function $a$ and impose $\kappa_\phi(x,y) = \kappa_\phi(x-y)$, we obtain that (3) is a convolution operator, which is a natural choice from the perspective of fundamental solutions. We exploit this fact in the following section by parameterizing $\kappa_\phi$ directly in Fourier space and using the Fast Fourier Transform (FFT) to efficiently compute (3). This leads to a fast architecture that obtains state-of-the-art results for PDE problems.

## 4 FOURIER NEURAL OPERATOR

We propose replacing the kernel integral operator in (3), by a convolution operator defined in Fourier space. Let $\mathcal{F}$ denote the Fourier transform of a function $f : D \to \mathbb{R}^{d_v}$ and $\mathcal{F}^{-1}$ its inverse then

$$(\mathcal{F}f)_j(k) = \int_D f_j(x)e^{-2i\pi\langle x,k\rangle}\mathrm{d}x, \qquad (\mathcal{F}^{-1}f)_j(x) = \int_D f_j(k)e^{2i\pi\langle x,k\rangle}\mathrm{d}k$$

for $j = 1,\ldots,d_v$ where $i = \sqrt{-1}$ is the imaginary unit. By letting $\kappa_\phi(x,y,a(x),a(y)) = \kappa_\phi(x-y)$ in (3) and applying the convolution theorem, we find that

$$\big(\mathcal{K}(a;\phi)v_t\big)(x) = \mathcal{F}^{-1}\big(\mathcal{F}(\kappa_\phi)\cdot\mathcal{F}(v_t)\big)(x), \qquad \forall x \in D.$$

We, therefore, propose to directly parameterize $\kappa_\phi$ in Fourier space.

**Definition 3 (Fourier integral operator $\mathcal{K}$)** *Define the Fourier integral operator*

$$\big(\mathcal{K}(\phi)v_t\big)(x) = \mathcal{F}^{-1}\Big(R_\phi \cdot (\mathcal{F}v_t)\Big)(x) \qquad \forall x \in D \tag{4}$$

*where* $R_\phi$ *is the Fourier transform of a periodic function* $\kappa : \bar{D} \to \mathbb{R}^{d_v \times d_v}$ *parameterized by* $\phi \in \Theta_\mathcal{K}$. *An illustration is given in Figure 2 (b).*

For frequency mode $k \in D$, we have $(\mathcal{F}v_t)(k) \in \mathbb{C}^{d_v}$ and $R_\phi(k) \in \mathbb{C}^{d_v \times d_v}$. Notice that since we assume $\kappa$ is periodic, it admits a Fourier series expansion, so we may work with the discrete modes $k \in \mathbb{Z}^d$. We pick a finite-dimensional parameterization by truncating the Fourier series at a maximal number of modes $k_{\max} = |Z_{k_{\max}}| = |\{k \in \mathbb{Z}^d : |k_j| \leq k_{\max,j}, \text{ for } j = 1,\ldots,d\}|$. We thus parameterize $R_\phi$ directly as complex-valued $(k_{\max} \times d_v \times d_v)$-tensor comprising a collection of truncated Fourier modes and therefore drop $\phi$ from our notation. Since $\kappa$ is real-valued, we impose conjugate symmetry. We note that the set $Z_{k_{\max}}$ is not the canonical choice for the low frequency modes of $v_t$. Indeed, the low frequency modes are usually defined by placing an upper-bound on the $\ell_1$-norm of $k \in \mathbb{Z}^d$. We choose $Z_{k_{\max}}$ as above since it allows for an efficient implementation.

**The discrete case and the FFT.** Assuming the domain $D$ is discretized with $n \in \mathbb{N}$ points, we have that $v_t \in \mathbb{R}^{n \times d_v}$ and $\mathcal{F}(v_t) \in \mathbb{C}^{n \times d_v}$. Since we convolve $v_t$ with a function which only has $k_{\max}$ Fourier modes, we may simply truncate the higher modes to obtain $\mathcal{F}(v_t) \in \mathbb{C}^{k_{\max} \times d_v}$. Multiplication by the weight tensor $R \in \mathbb{C}^{k_{\max} \times d_v \times d_v}$ is then

$$\big(R \cdot (\mathcal{F}v_t)\big)_{k,l} = \sum_{j=1}^{d_v} R_{k,l,j}(\mathcal{F}v_t)_{k,j}, \qquad k = 1,\ldots,k_{\max}, \quad j = 1,\ldots,d_v. \tag{5}$$

When the discretization is uniform with resolution $s_1 \times \cdots \times s_d = n$, $\mathcal{F}$ can be replaced by the Fast Fourier Transform. For $f \in \mathbb{R}^{n \times d_v}$, $k = (k_1,\ldots,k_d) \in \mathbb{Z}_{s_1} \times \cdots \times \mathbb{Z}_{s_d}$, and $x = (x_1,\ldots,x_d) \in D$, the FFT $\hat{\mathcal{F}}$ and its inverse $\hat{\mathcal{F}}^{-1}$ are defined as

$$(\hat{\mathcal{F}}f)_l(k) = \sum_{x_1=0}^{s_1-1} \cdots \sum_{x_d=0}^{s_d-1} f_l(x_1,\ldots,x_d)e^{-2i\pi\sum_{j=1}^d \frac{x_j k_j}{s_j}},$$

$$(\hat{\mathcal{F}}^{-1}f)_l(x) = \sum_{k_1=0}^{s_1-1} \cdots \sum_{k_d=0}^{s_d-1} f_l(k_1,\ldots,k_d)e^{2i\pi\sum_{j=1}^d \frac{x_j k_j}{s_j}}$$

for $l = 1, \ldots, d_v$. In this case, the set of truncated modes becomes

$$Z_{k_{\max}} = \{(k_1, \ldots, k_d) \in \mathbb{Z}_{s_1} \times \cdots \times \mathbb{Z}_{s_d} \mid k_j \leq k_{\max,j} \text{ or } s_j - k_j \leq k_{\max,j}, \text{ for } j = 1, \ldots, d\}.$$

When implemented, $R$ is treated as a $(s_1 \times \cdots \times s_d \times d_v \times d_v)$-tensor and the above definition of $Z_{k_{\max}}$ corresponds to the "corners" of $R$, which allows for a straight-forward parallel implementation of (5) via matrix-vector multiplication. In practice, we have found that choosing $k_{\max,j} = 12$ which yields $k_{\max} = 12^d$ parameters per channel to be sufficient for all the tasks that we consider.

**Parameterizations of $R$.** In general, $R$ can be defined to depend on $(\mathcal{F}a)$ to parallel (3). Indeed, we can define $R_\phi : \mathbb{Z}^d \times \mathbb{R}^{d_v} \to \mathbb{R}^{d_v \times d_v}$ as a parametric function that maps $\big(k, (\mathcal{F}a)(k)\big)$ to the values of the appropriate Fourier modes. We have experimented with linear as well as neural network parameterizations of $R_\phi$. We find that the linear parameterization has a similar performance to the previously described direct parameterization, while neural networks have worse performance. This is likely due to the discrete structure of the space $\mathbb{Z}^d$. Our experiments in this work focus on the direct parameterization presented above.

**Invariance to discretization.** The Fourier layers are discretization-invariant because they can learn from and evaluate functions which are discretized in an arbitrary way. Since parameters are learned directly in Fourier space, resolving the functions in physical space simply amounts to projecting on the basis $e^{2\pi i \langle x, k \rangle}$ which are well-defined everywhere on $\mathbb{R}^d$. This allows us to achieve zero-shot super-resolution as shown in Section 5.4. Furthermore, our architecture has a consistent error at any resolution of the inputs and outputs. On the other hand, notice that, in Figure 3, the standard CNN methods we compare against have an error that grows with the resolution.

**Quasi-linear complexity.** The weight tensor $R$ contains $k_{\max} < n$ modes, so the inner multiplication has complexity $O(k_{\max})$. Therefore, the majority of the computational cost lies in computing the Fourier transform $\mathcal{F}(v_t)$ and its inverse. General Fourier transforms have complexity $O(n^2)$, however, since we truncate the series the complexity is in fact $O(nk_{\max})$, while the FFT has complexity $O(n \log n)$. Generally, we have found using FFTs to be very efficient. However a uniform discretization is required.

## 5 NUMERICAL EXPERIMENTS

In this section, we compare the proposed Fourier neural operator with multiple finite-dimensional architectures as well as operator-based approximation methods on the 1-d Burgers' equation, the 2-d Darcy Flow problem, and 2-d Navier-Stokes equation. The data generation processes are discussed in Appendices A.3.1, A.3.2, and A.3.3 respectively. We do not compare against traditional solvers (FEM/FDM) or neural-FEM type methods since our goal is to produce an efficient operator approximation that can be used for downstream applications. We demonstrate one such application to the Bayesian inverse problem in Section 5.5.

We construct our Fourier neural operator by stacking four Fourier integral operator layers as specified in (2) and (4) with the ReLU activation as well as batch normalization. Unless otherwise specified, we use $N = 1000$ training instances and 200 testing instances. We use Adam optimizer to train for 500 epochs with an initial learning rate of 0.001 that is halved every 100 epochs. We set $k_{\max,j} = 16, d_v = 64$ for the 1-d problem and $k_{\max,j} = 12, d_v = 32$ for the 2-d problems. Lower resolution data are downsampled from higher resolution. All the computation is carried on a single Nvidia V100 GPU with 16GB memory.

**Remark on Resolution.** Traditional PDE solvers such as FEM and FDM approximate a single function and therefore their error to the continuum decreases as the resolution is increased. On the other hand, operator approximation is independent of the ways its data is discretized as long as all relevant information is resolved. Resolution-invariant operators have consistent error rates among different resolutions as shown in Figure 3. Further, resolution-invariant operators can do zero-shot super-resolution, as shown in Section 5.4.

**Benchmarks for time-independent problems (Burgers and Darcy):** **NN:** a simple point-wise feedforward neural network. **RBM:** the classical Reduced Basis Method (using a POD basis) (De-

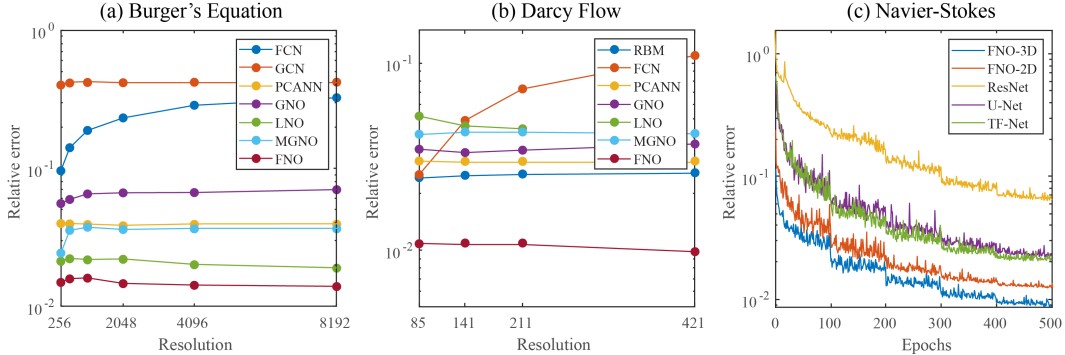

**Left:** benchmarks on Burgers equation; **Mid:** benchmarks on Darcy Flow for different resolutions; **Right:** the learning curves on Navier-Stokes $\nu = 1\mathrm{e}{-3}$ with different benchmarks. Train and test on the same resolution. For acronyms, see Section 5; details in Tables 1, 3, 4.

Figure 3: Benchmark on Burger's equation, Darcy Flow, and Navier-Stokes

Vore, 2014). **FCN:** a the-state-of-the-art neural network architecture based on Fully Convolution Networks (Zhu & Zabaras, 2018). **PCANN:** an operator method using PCA as an autoencoder on both the input and output data and interpolating the latent spaces with a neural network (Bhattacharya et al., 2020). **GNO:** the original graph neural operator (Li et al., 2020b). **MGNO:** the multipole graph neural operator (Li et al., 2020a). **LNO:** a neural operator method based on the low-rank decomposition of the kernel $\kappa(x, y) := \sum_{j=1}^{r} \phi_j(x)\psi_j(y)$, similar to the unstacked Deep-ONet proposed in (Lu et al., 2019). **FNO:** the newly purposed Fourier neural operator.

**Benchmarks for time-dependent problems (Navier-Stokes):** **ResNet:** 18 layers of 2-d convolution with residual connections (He et al., 2016). **U-Net:** A popular choice for image-to-image regression tasks consisting of four blocks with 2-d convolutions and deconvolutions (Ronneberger et al., 2015). **TF-Net:** A network designed for learning turbulent flows based on a combination of spatial and temporal convolutions (Wang et al., 2020). **FNO-2d:** 2-d Fourier neural operator with a RNN structure in time. **FNO-3d:** 3-d Fourier neural operator that directly convolves in space-time.

## 5.1 BURGERS' EQUATION

The 1-d Burgers' equation is a non-linear PDE with various applications including modeling the one dimensional flow of a viscous fluid. It takes the form

$$\partial_t u(x,t) + \partial_x(u^2(x,t)/2) = \nu \partial_{xx} u(x,t), \qquad x \in (0,1), t \in (0,1] \tag{6}$$
$$u(x,0) = u_0(x), \qquad x \in (0,1)$$

with periodic boundary conditions where $u_0 \in L^2_{\mathrm{per}}((0,1); \mathbb{R})$ is the initial condition and $\nu \in \mathbb{R}_+$ is the viscosity coefficient. We aim to learn the operator mapping the initial condition to the solution at time one, $G^\dagger : L^2_{\mathrm{per}}((0,1); \mathbb{R}) \to H^r_{\mathrm{per}}((0,1); \mathbb{R})$ defined by $u_0 \mapsto u(\cdot, 1)$ for any $r > 0$.

The results of our experiments are shown in Figure 3 (a) and Table 3 (Appendix A.3.1). Our proposed method obtains the lowest relative error compared to any of the benchmarks. Further, the error is invariant with the resolution, while the error of convolution neural network based methods (FCN) grows with the resolution. Compared to other neural operator methods such as GNO and MGNO that use Nyström sampling in physical space, the Fourier neural operator is both more accurate and more computationally efficient.

## 5.2 DARCY FLOW

We consider the steady-state of the 2-d Darcy Flow equation on the unit box which is the second order, linear, elliptic PDE

$$-\nabla \cdot (a(x)\nabla u(x)) = f(x) \qquad x \in (0,1)^2 \tag{7}$$
$$u(x) = 0 \qquad x \in \partial(0,1)^2$$

with a Dirichlet boundary where $a \in L^{\infty}((0,1)^2; \mathbb{R}_+)$ is the diffusion coefficient and $f \in L^2((0,1)^2; \mathbb{R})$ is the forcing function. This PDE has numerous applications including modeling the pressure of subsurface flow, the deformation of linearly elastic materials, and the electric potential in conductive materials. We are interested in learning the operator mapping the diffusion coefficient to the solution, $G^{\dagger} : L^{\infty}((0,1)^2; \mathbb{R}_+) \rightarrow H_0^1((0,1)^2; \mathbb{R}_+)$ defined by $a \mapsto u$. Note that although the PDE is linear, the operator $G^{\dagger}$ is not.

The results of our experiments are shown in Figure 3 (b) and Table 4 (Appendix A.3.2). The proposed Fourier neural operator obtains nearly one order of magnitude lower relative error compared to any benchmarks. We again observe the invariance of the error with respect to the resolution.

## 5.3 NAVIER-STOKES EQUATION

We consider the 2-d Navier-Stokes equation for a viscous, incompressible fluid in vorticity form on the unit torus:

$$
\begin{aligned}
\partial_t w(x,t) + u(x,t) \cdot \nabla w(x,t) &= \nu \Delta w(x,t) + f(x), & x \in (0,1)^2, t \in (0,T] \\
\nabla \cdot u(x,t) &= 0, & x \in (0,1)^2, t \in [0,T] \\
w(x,0) &= w_0(x), & x \in (0,1)^2
\end{aligned}
\tag{8}
$$

where $u \in C([0,T]; H_{\text{per}}^r((0,1)^2; \mathbb{R}^2))$ for any $r > 0$ is the velocity field, $w = \nabla \times u$ is the vorticity, $w_0 \in L_{\text{per}}^2((0,1)^2; \mathbb{R})$ is the initial vorticity, $\nu \in \mathbb{R}_+$ is the viscosity coefficient, and $f \in L_{\text{per}}^2((0,1)^2; \mathbb{R})$ is the forcing function. We are interested in learning the operator mapping the vorticity up to time 10 to the vorticity up to some later time $T > 10$, $G^{\dagger} : C([0,10]; H_{\text{per}}^r((0,1)^2; \mathbb{R})) \rightarrow C((10,T]; H_{\text{per}}^r((0,1)^2; \mathbb{R}))$ defined by $w|_{(0,1)^2 \times [0,10]} \mapsto w|_{(0,1)^2 \times (10,T]}$. Given the vorticity it is easy to derive the velocity. While vorticity is harder to model compared to velocity, it provides more information. By formulating the problem on vorticity, the neural network models mimic the pseudo-spectral method. We experiment with the viscosities $\nu = 1e{-}3, 1e{-}4, 1e{-}5$, decreasing the final time $T$ as the dynamic becomes chaotic. Since the baseline methods are not resolution-invariant, we fix the resolution to be $64 \times 64$ for both training and testing.

Table 1: Benchmarks on Navier Stokes (fixing resolution $64 \times 64$ for both training and testing)

| Config | Parameters | Time per epoch | $\nu = 1e{-}3$ $T = 50$ $N = 1000$ | $\nu = 1e{-}4$ $T = 30$ $N = 1000$ | $\nu = 1e{-}4$ $T = 30$ $N = 10000$ | $\nu = 1e{-}5$ $T = 20$ $N = 1000$ |
|---|---|---|---|---|---|---|
| FNO-3D | $6,558,537$ | $38.99s$ | **0.0086** | 0.1918 | **0.0820** | 0.1893 |
| FNO-2D | $414,517$ | $127.80s$ | 0.0128 | **0.1559** | 0.0834 | **0.1556** |
| U-Net | $24,950,491$ | $48.67s$ | 0.0245 | 0.2051 | 0.1190 | 0.1982 |
| TF-Net | $7,451,724$ | $47.21s$ | 0.0225 | 0.2253 | 0.1168 | 0.2268 |
| ResNet | $266,641$ | $78.47s$ | 0.0701 | 0.2871 | 0.2311 | 0.2753 |

As shown in Table 1, the FNO-3D has the best performance when there is sufficient data ($\nu = 1e{-}3, N = 1000$ and $\nu = 1e{-}4, N = 10000$). For the configurations where the amount of data is insufficient ($\nu = 1e{-}4, N = 1000$ and $\nu = 1e{-}5, N = 1000$), all methods have $> 15\%$ error with FNO-2D achieving the lowest. Note that we only present results for spatial resolution $64 \times 64$ since all benchmarks we compare against are designed for this resolution. Increasing it degrades their performance while FNO achieves the same errors.

**2D and 3D Convolutions.** FNO-2D, U-Net, TF-Net, and ResNet all do 2D-convolution in the spatial domain and recurrently propagate in the time domain (2D+RNN). The operator maps the solution at the previous 10 time steps to the next time step (2D functions to 2D functions). On the other hand, FNO-3D performs convolution in space-time. It maps the initial time steps directly to the full trajectory (3D functions to 3D functions). The 2D+RNN structure can propagate the solution to any arbitrary time $T$ in increments of a fixed interval length $\Delta t$, while the Conv3D structure is fixed to the interval $[0,T]$ but can transfer the solution to an arbitrary time-discretization. We find the 3-d method to be more expressive and easier to train compared to its RNN-structured counterpart.

### 5.4 ZERO-SHOT SUPER-RESOLUTION.

The neural operator is mesh-invariant, so it can be trained on a lower resolution and evaluated at a higher resolution, without seeing any higher resolution data (zero-shot super-resolution). Figure 1 shows an example where we train the FNO-3D model on $64 \times 64 \times 20$ resolution data in the setting above with ($\nu = 1e{-}4, N = 10000$) and transfer to $256 \times 256 \times 80$ resolution, demonstrating super-resolution in space-time. Fourier neural operator is the only model among the benchmarks (FNO-2D, U-Net, TF-Net, and ResNet) that can do zero-shot super-resolution. And surprisingly, it can do super-resolution not only in the spatial domain but also in the temporal domain.

### 5.5 BAYESIAN INVERSE PROBLEM

In this experiment, we use a function space Markov chain Monte Carlo (MCMC) method (Cotter et al., 2013) to draw samples from the posterior distribution of the initial vorticity in Navier-Stokes given sparse, noisy observations at time $T = 50$. We compare the Fourier neural operator acting as a surrogate model with the traditional solvers used to generate our train-test data (both run on GPU). We generate 25,000 samples from the posterior (with a 5,000 sample burn-in period), requiring 30,000 evaluations of the forward operator.

As shown in Figure 6 (Appendix A.5), FNO and the traditional solver recover almost the same posterior mean which, when pushed forward, recovers well the late-time dynamic of Navier Stokes. In sharp contrast, FNO takes $0.005s$ to evaluate a single instance while the traditional solver, after being optimized to use the largest possible internal time-step which does not lead to blow-up, takes $2.2s$. This amounts to 2.5 minutes for the MCMC using FNO and over 18 hours for the traditional solver. Even if we account for data generation and training time (offline steps) which take 12 hours, using FNO is still faster! Once trained, FNO can be used to quickly perform multiple MCMC runs for different initial conditions and observations, while the traditional solver will take 18 hours for every instance. Furthermore, since FNO is differentiable, it can easily be applied to PDE-constrained optimization problems without the need for the adjoint method.

**Spectral analysis.** Due to the way we parameterize $R_\phi$, the function output by (4) has at most $k_{\max,j}$ Fourier modes per channel. This, however, does not mean that the Fourier neural operator can only approximate functions up to $k_{\max,j}$ modes. Indeed, the activation functions which occur between integral operators and the final decoder network $Q$ recover the high frequency modes. As an example, consider a solution to the Navier-Stokes equation with viscosity $\nu = 1e{-}3$. Truncating this function at 20 Fourier modes yields an error around 2% while our Fourier neural operator learns the parametric dependence and produces approximations to an error of $\leq 1\%$ with only $k_{\max,j} = 12$ parameterized modes.

**Non-periodic boundary condition.** Traditional Fourier methods work only with periodic boundary conditions. However, the Fourier neural operator does not have this limitation. This is due to the linear transform $W$ (the bias term) which keeps the track of non-periodic boundary. As an example, the Darcy Flow and the time domain of Navier-Stokes have non-periodic boundary conditions, and the Fourier neural operator still learns the solution operator with excellent accuracy.

## 6 DISCUSSION AND CONCLUSION

**Requirements on Data.** Data-driven methods rely on the quality and quantity of data. To learn Navier-Stokes equation with Reynolds number $Re = 1e{+}4$, we need to generate $N = 10000$ training pairs $\{a_j, u_j\}$ with the numerical solver. However, for more challenging PDEs, generating a few training samples can be already very expensive. A future direction is to combine neural operators with numerical solvers to levitate the requirements on data. **Recurrent structure.** The neural operator has an iterative structure that can naturally be formulated as a recurrent network where all layers share the same parameters without sacrificing performance. (We did not impose this restriction in the experiments.) **Computer vision.** Operator learning is not restricted to PDEs. Images can naturally be viewed as real-valued functions on 2-d domains and videos simply add a temporal structure. Our approach is therefore a natural choice for problems in computer vision where invariance to discretization crucial is important (Chi et al., 2020).

## ACKNOWLEDGEMENTS

The authors want to thank Ray Wang and Rose Yu for meaningful discussions. Z. Li gratefully acknowledges the financial support from the Kortschak Scholars Program. A. Anandkumar is supported in part by Bren endowed chair, LwLL grants, Beyond Limits, Raytheon, Microsoft, Google, Adobe faculty fellowships, and DE Logi grant. K. Bhattacharya, N. B. Kovachki, B. Liu, and A. M. Stuart gratefully acknowledge the financial support of the Army Research Laboratory through the Cooperative Agreement Number W911NF-12-0022. Research was sponsored by the Army Research Laboratory and was accomplished under Cooperative Agreement Number W911NF-12-2-0022. The views and conclusions contained in this document are those of the authors and should not be interpreted as representing the official policies, either expressed or implied, of the Army Research Laboratory or the U.S. Government. The U.S. Government is authorized to reproduce and distribute reprints for Government purposes notwithstanding any copyright notation herein.

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

# A  APPENDIX

## A.1  TABLE OF NOTATIONS

A table of notations is given in Table 2.

Table 2: table of notations

| Notation | Meaning |
|---|---|
| **Operator learning** | |
| $D \subset \mathbb{R}^d$ | The spatial domain for the PDE |
| $x \in D$ | Points in the the spatial domain |
| $a \in \mathcal{A} = (D; \mathbb{R}^{d_a})$ | The input coefficient functions |
| $u \in \mathcal{U} = (D; \mathbb{R}^{d_u})$ | The target solution functions |
| $D_j$ | The discretization of $(a_j, u_j)$ |
| $G^\dagger : \mathcal{A} \to \mathcal{U}$ | The operator mapping the coefficients to the solutions |
| $\mu$ | A probability measure where $a_j$ sampled from. |
| **Neural operator** | |
| $v(x) \in \mathbb{R}^{d_v}$ | The neural network representation of $u(x)$ |
| $d_a$ | Dimension of the input $a(x)$. |
| $d_u$ | Dimension of the output $u(x)$. |
| $d_v$ | The dimension of the representation $v(x)$ |
| $\kappa : \mathbb{R}^{2(d+1)} \to \mathbb{R}^{d_v \times d_v}$ | The kernel maps $(x, y, a(x), a(y))$ to a $d_v \times d_v$ matrix |
| $\phi$ | The parameters of the kernel network $\kappa$ |
| $t = 0, \ldots, T$ | The time steps (layers) |
| $\sigma$ | The activation function |
| **Fourier operator** | |
| $\mathcal{F}, \mathcal{F}^{-1}$ | Fourier transformation and its inverse. |
| $R$ | The linear transformation applied on the lower Fourier modes. |
| $W$ | The linear transformation (bias term) applied on the spatial domain. |
| $k$ | Fourier modes / wave numbers. |
| $k_{max}$ | The max Fourier modes used in the Fourier layer. |
| **Hyperparameters** | |
| $N$ | The number of training pairs. |
| $n$ | The size of the discretization. |
| $s$ | The resolution of the discretization ($s^d = n$). |
| $\nu$ | The viscosity. |
| $T$ | The time interval $[0, T]$ for time-dependent equation. |

## A.2  SPECTRAL ANALYSIS

The spectral decay of the Navier Stokes equation data is shown in Figure 4. The spectrum decay has a slope $k^{-5/3}$, matching the energy spectrum in the turbulence region. And we notice the energy spectrum does not decay along with time.

We also present the spectral decay in term of the truncation $k_{max}$ defined in 4 as shown in Figure5. We note all equations (Burgers, Darcy, and Navier-Stokes with $\nu \leq 1e-4$ ) exhibit high frequency modes. Even we truncate at $k_{max} = 12$ in the Fourier layer, the Fourier neural operator can recover the high frequency modes.

## A.3  DATA GENERATION

In this section, we provide the details of data generator for the three equation we used in Section 5.

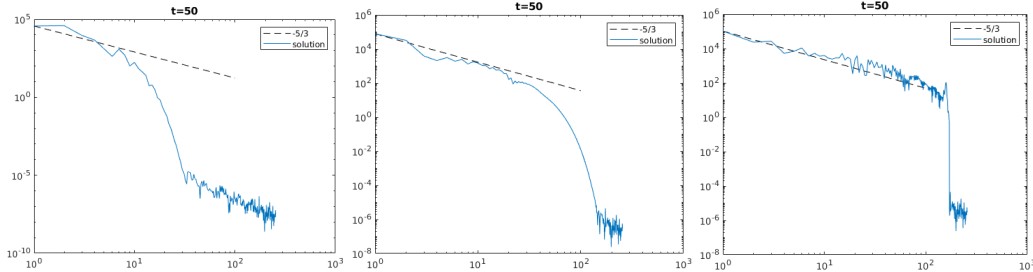

The spectral decay of the Navier-stokes equation data we used in section 5.3. The y-axis is the spectrum; the x-axis is the wavenumber $|k| = k_1 + k_2$.

Figure 4: Spectral Decay of Navier-Stokes equations

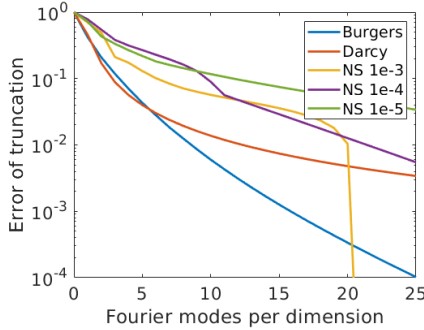

The error of truncation in one single Fourier layer without applying the linear transform $R$. The y-axis is the normalized truncation error; the x-axis is the truncation mode $k_{max}$.

Figure 5: Spectral Decay in term of $k_{max}$

### A.3.1   BURGERS EQUATION

Recall the 1-d Burger's equation on the unit torus:

$$\partial_t u(x,t) + \partial_x(u^2(x,t)/2) = \nu \partial_{xx} u(x,t), \qquad x \in (0,1), t \in (0,1]$$
$$u(x,0) = u_0(x), \qquad x \in (0,1).$$

The initial condition $u_0(x)$ is generated according to $u_0 \sim \mu$ where $\mu = \mathcal{N}(0, 625(-\Delta + 25I)^{-2})$ with periodic boundary conditions. We set the viscosity to $\nu = 0.1$ and solve the equation using a split step method where the heat equation part is solved exactly in Fourier space then the non-linear part is advanced, again in Fourier space, using a very fine forward Euler method. We solve on a spatial mesh with resolution $2^{13} = 8192$ and use this dataset to subsample other resolutions.

### A.3.2   DARCY FLOW

The 2-d Darcy Flow is a second-order linear elliptic equation of the form

$$-\nabla \cdot (a(x)\nabla u(x)) = f(x) \qquad x \in (0,1)^2$$
$$u(x) = 0 \qquad x \in \partial(0,1)^2.$$

The coefficients $a(x)$ are generated according to $a \sim \mu$ where $\mu = \psi_\# \mathcal{N}(0, (-\Delta + 9I)^{-2})$ with zero Neumann boundary conditions on the Laplacian. The mapping $\psi : \mathbb{R} \to \mathbb{R}$ takes the value 12 on the positive part of the real line and 3 on the negative and the push-forward is defined pointwise. The forcing is kept fixed $f(x) = 1$. Such constructions are prototypical models for many physical systems such as permeability in subsurface flows and material microstructures in elasticity. Solutions $u$ are obtained by using a second-order finite difference scheme on a $421 \times 421$ grid. Different resolutions are downsampled from this dataset.

### A.3.3 NAVIER-STOKES EQUATION

Recall the 2-d Navier-Stokes equation for a viscous, incompressible fluid in vorticity form on the unit torus:

$$\partial_t w(x,t) + u(x,t) \cdot \nabla w(x,t) = \nu \Delta w(x,t) + f(x), \qquad x \in (0,1)^2, t \in (0,T]$$
$$\nabla \cdot u(x,t) = 0, \qquad x \in (0,1)^2, t \in [0,T]$$
$$w(x,0) = w_0(x), \qquad x \in (0,1)^2.$$

The initial condition $w_0(x)$ is generated according to $w_0 \sim \mu$ where $\mu = \mathcal{N}(0, 7^{3/2}(-\Delta+49I)^{-2.5})$ with periodic boundary conditions. The forcing is kept fixed $f(x) = 0.1(\sin(2\pi(x_1 + x_2)) + \cos(2\pi(x_1 + x_2)))$. The equation is solved using the stream-function formulation with a pseudospectral method. First a Poisson equation is solved in Fourier space to find the velocity field. Then the vorticity is differentiated and the non-linear term is computed is physical space after which it is dealiased. Time is advanced with a Crank–Nicolson update where the non-linear term does not enter the implicit part. All data are generated on a $256 \times 256$ grid and are downsampled to $64 \times 64$. We use a time-step of $1e{-}4$ for the Crank–Nicolson scheme in the data-generated process where we record the solution every $t = 1$ time units. The step is increased to $2e{-}2$ when used in MCMC for the Bayesian inverse problem.

### A.4 RESULTS OF BURGERS' EQUATION AND DARCY FLOW

The details error rate on Burgers' equation and Darcy Flow are listed in Table 3 and Table 4.

Table 3: Benchmarks on 1-d Burgers' equation

| Networks | $s = 256$ | $s = 512$ | $s = 1024$ | $s = 2048$ | $s = 4096$ | $s = 8192$ |
|---|---|---|---|---|---|---|
| NN | 0.4714 | 0.4561 | 0.4803 | 0.4645 | 0.4779 | 0.4452 |
| GCN | 0.3999 | 0.4138 | 0.4176 | 0.4157 | 0.4191 | 0.4198 |
| FCN | 0.0958 | 0.1407 | 0.1877 | 0.2313 | 0.2855 | 0.3238 |
| PCANN | 0.0398 | 0.0395 | 0.0391 | 0.0383 | 0.0392 | 0.0393 |
| GNO | 0.0555 | 0.0594 | 0.0651 | 0.0663 | 0.0666 | 0.0699 |
| LNO | 0.0212 | 0.0221 | 0.0217 | 0.0219 | 0.0200 | 0.0189 |
| MGNO | 0.0243 | 0.0355 | 0.0374 | 0.0360 | 0.0364 | 0.0364 |
| FNO | **0.0149** | **0.0158** | **0.0160** | **0.0146** | **0.0142** | **0.0139** |

Table 4: Benchmarks on 2-d Darcy Flow

| Networks | $s = 85$ | $s = 141$ | $s = 211$ | $s = 421$ |
|---|---|---|---|---|
| NN | 0.1716 | 0.1716 | 0.1716 | 0.1716 |
| FCN | 0.0253 | 0.0493 | 0.0727 | 0.1097 |
| PCANN | 0.0299 | 0.0298 | 0.0298 | 0.0299 |
| RBM | 0.0244 | 0.0251 | 0.0255 | 0.0259 |
| GNO | 0.0346 | 0.0332 | 0.0342 | 0.0369 |
| LNO | 0.0520 | 0.0461 | 0.0445 | − |
| MGNO | 0.0416 | 0.0428 | 0.0428 | 0.0420 |
| FNO | **0.0108** | **0.0109** | **0.0109** | **0.0098** |

### A.5 BAYESIAN INVERSE PROBLEM

Results of the Bayesian inverse problem for the Navier-Stokes equation are shown in Figure 6. It can be seen that the result using Fourier neural operator as a surrogate is as good as the result of the traditional solver.

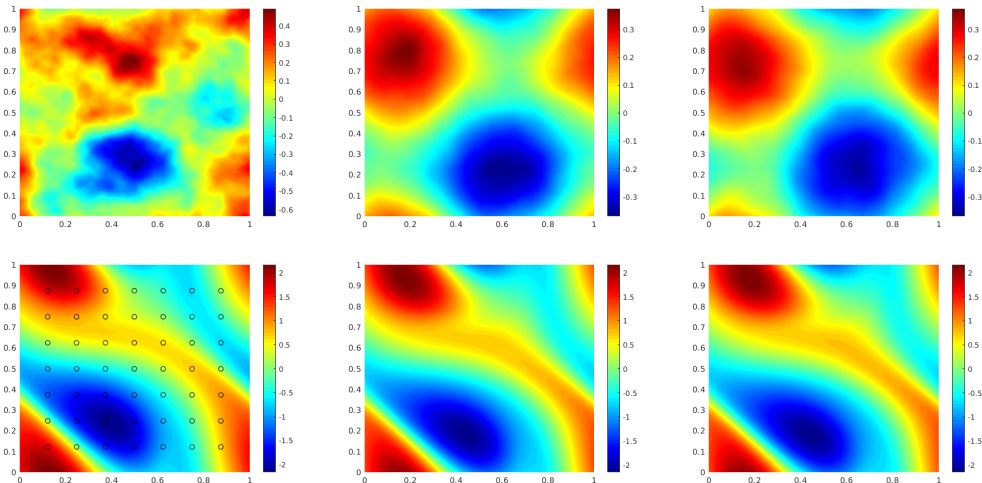

The top left panel shows the true initial vorticity while bottom left panel shows the true observed vorticity at $T = 50$ with black dots indicating the locations of the observation points placed on a $7 \times 7$ grid. The top middle panel shows the posterior mean of the initial vorticity given the noisy observations estimated with MCMC using the traditional solver, while the top right panel shows the same thing but using FNO as a surrogate model. The bottom middle and right panels show the vorticity at $T = 50$ when the respective approximate posterior means are used as initial conditions.

Figure 6: Results of the Bayesian inverse problem for the Navier-Stokes equation.

