# OpenReview forum: "Fourier Neural Operator for Parametric Partial Differential Equations"
_ICLR.cc/2021/Conference — ICLR 2021 Poster_

### Official Review · AnonReviewer4 · 2020-10-28
**I vote for reject. However,  I think that the work could be published after addressing some concerns.**

**Rating:** 5
**Confidence:** 5

**Review:**

The manuscript proposes a neural network for solving partial differential equations. This is performed by a supervised learning of a mapping between the PDE coefficients (a) and the PDE solution (u), where a is drawn from a certain distribution. This work follows  a work by Li et al. 2020b.  Inspired by the properties of Green’s function, in Li et al. the authors suggest to learn a kernel function in an iterative manner.  In this work, the general kernel is replaced by a convolution kernel which is represented in the Fourier space.




Pros.

1.	The authors address an important and practical problem
2.	I like the idea of learning a mapping for a class of PDEs, rather than optimizing per instance
3.	The numerical results are impressive

Cons.
1.	Novelty. I feel that the novelty is incremental with respect to Li et al. The motivation for replacing the kernel function (3) by a convolution operator is not entirely clear. I guess that efficiency is one of the reasons. If so, I would suggest to discuss the complexity of the proposed scheme vs. Li et al. complexity.  Moreover, how does it affect the expressivity of the network?
2.	Clarity of the paper.  I didn’t like the idea that on the one hand entire sentences are taken form Li et al. but on the other hand, the paper itself is not self-contained. In practice, the reader should follow the intuition and the motivation given in Li et al in order to grasp the idea of neural operator.  In addition, a concise detailed description of the network architecture is missing.
3.	Missing references and comparison.  There are two works, aiming at solving PDEs, which are very relevant in the context of the proposed approach. The first copes with the structured case, Greenfeld at al. “Learning to optimize multigrid PDE solvers” and the second handles the unstructured case, Luz et al. “Learning algebraic multigrid using graph neural networks”. It resembles the proposed approach in the sense that the training is performed for a class of problems, yielding a mapping from the coefficients to the solution. In these works, the learning is unsupervised, while generalizing over boundary conditions and domains.
To summarize, I believe that this work should be published. I hope that the authors are able to address my concerns.

---

> ### Author Response · Authors · 2020-11-23
> **Response to Reviewer 4**
>
> We want to thank reviewer 4 for these valuable opinions. Regarding the questions:
>
> 1. **Novelty and improvement**: We develop a novel neural operator framework based on Fourier transform, substantially different from the previous graph-based neural operators. Our method has better complexity and efficiency. Fourier neural operator is the first work that learns the resolution-invariant solution operator for the family of Navier-Stokes equations in the turbulent regime. Previous graph-based neural operators do not converge on the challenging cases such as Navier-Stokes.
>     - **Complexity**: Our model has quasilinear complexity while the original neural operator is quadratic. In practice, our model is much faster than the previous graph-based models. It is also up to three orders of magnitude faster than the conventional pseudo-spectral solver.
>     - **Accuracy**: We observe Fourier filters are more efficient to represent continuous functions in PDEs. It achieves error rates that are 30% lower on Burgers’ Equation, 60% lower on Darcy Flow, and 30% lower on Navier Stokes.
>     - **Expressivity**: We observe restricting the integration to a convolution does not compromise the expressiveness of neural operators. We empirically show that the proposed architecture can approximate complicated operators arising in PDEs that are highly non-linear, with high frequency modes, and slow spectral decay:
>         1. **Non-linearity**: The power of the network comes from combining linear integral operators (via the Fourier transform) and non-linear activation functions. As a result it can learn highly non-linear operators. This is analogous to the standard MLP and CNN where we combine linear multiplications with non-linear activations which allows us to learn highly nonlinear functions.
>         2. **High frequency modes**: Although we truncate out the higher frequency modes in the Fourier layer, the operator as a whole can approximate functions with the full frequency range. This is because we represent the function in a high dimensional channel space. When projecting back to the desired dimension, the non-linear decoder network recovers the higher frequency modes.
>         3. **Non-periodic boundary**: The Fourier layer can only approximate functions with periodic boundary, but it is not a limitation of the operator as a whole. The linear transform W outside the Fourier transform (the bias term) keeps track of the non-periodic, location information. Therefore, the full architecture can learn solution functions with non-periodic boundaries.
> 2. **Clarity**: Thanks for pointing this out. We have added a new diagram in the revised paper to explain the full architecture, and we have updated the paper to make it more self-consistent.
> 3. **Reference**: Thanks for the references. We are indeed familiar with these works. In these works, the authors use neural networks to learn prolongation matrices to enhance multigrid solvers. This framework enhances multigrid solvers, but it does not fit into the operator learning setting. Given a new query, one still needs to run the solver and iterate the V-cycle algorithm with the learned prolongation matrix. On the other hand, the neural operator directly learns the solution operators. And given a new query, one can immediately get the evaluation. Although we cannot directly compare with these works, we do have benchmarks with Graph neural network-based operator (GNO) and Multipole graph neural network-based operator (MGNO) which are similar to the frameworks in the reference. We have added a reference to the paper.

---

### Official Review · AnonReviewer1 · 2020-10-29
**Review of "Fourier Neural Operator for Parametric Partial Differential Equations"**

**Rating:** 8
**Confidence:** 4

**Review:**

Paper summary:

Building on previous work on neural operators, the paper introduces the Fourier neural operator, which uses a convolution operator defined in Fourier space in place of the usual kernel integral operator. Each step of the neural operator then amounts to applying a Fourier transform to a vector (or rather, a set of vectors on a mesh), performing a linear transform (learnt parameters in this model) on the transformed vector, before performing an inverse Fourier transform on the result, recombining it with a linear map of the original vector, and passing the total result through a non-linearity. The Fourier neural operator is by construction (like all neural operators) a map between function spaces, and invariance to discretization follows immediately from the nature of a Fourier transform (just project onto the usual basis). If the underlying domain has a uniform discretization, the fast Fourier transformation (FFT) can be used, allowing for an O(nlogn) evaluation of the aforementioned convolution operator, where n is the number of points in the discretization. Experiments demonstrate that the Fourier neural operator significantly outperforms other neural operators and other deep learning methods on Burgers’ equation, Darcy Flow, and Navier Stokes, and that that it is also significantly faster than traditional PDE solvers.

------------------------------------------
Strengths and weaknesses:

Much of the theoretical legwork for this paper, namely, neural operators, was already carried out in previous papers (Li et al.). The remaining theoretical work, namely writing down the Fourier integral operator and analysing the discrete case, was succinctly explained. The subsequent experimentation was extremely thorough (e.g. demonstrating that activation functions help in recovering high frequency modes) and, of course, the results were very impressive. I liked the paper a lot, and it’s definitely a big step-forward in neural operators. I’m assigning a score of 8 (a very good conference paper), and I think that the paper is more or less ready for publication as is. I’ve included a few questions below (to help my own understanding), as well as some typos I spotted whilst reading the paper.

------------------------------------------
Questions and clarification requests:

1)	Section 4, The Discrete Case and the FFT – could you explain the definition of bounds in the definition of Z_{k_{max}}?
2)	Section 4, Parametrizations of R, sentence 2 – could you explain the definition R_{\phi}? At present I can’t see how the function signature of R matches the definition given.
------------------------------------------
Typos and minor edits:
- Page 3, bullet point 3 – “solving Bayesian inference problem” -> “solving Bayesian inference problems”
- Section 1, final paragraph, sentence 2 - “approximate function with any boundary conditions” -> “approximate functions with any boundary conditions”
- Section 4, The discrete case and the FFT, final paragraph, last sentence - “all the task that we consider” -> “all the tasks that we consider”
- Section 4, Parametrizations of R, last sentence - “while neural networks have the worse performance” -> “while neural networks have the worst performance”
- Section 4, final sentence – “Generally, we have found using FFTs to be very efficient, however a uniform discretization if required.” -> “Generally, we have found using FFTs to be very efficient. However, a uniform discretization is required.”
- Section 5, final paragraph, sentence 2 – “FNO takes 0.005s to evaluate a single instances while the traditional solver” -> “FNO takes 0.005s to evaluate a single instance while the traditional solver”
- Section 6, final sentence – “Traditional Fourier methods work only with periodic boundary conditions, however, our Fourier neural operator does not have this limitation.” -> “Traditional Fourier methods work only with periodic boundary conditions. However, our Fourier neural operator does not have this limitation.”

---

> ### Author Response · Authors · 2020-11-23
> **Response to Reviewer 1**
>
> Thank reviewer 1 for the appreciation of our work! For the questions:
> 1. **Definition of $Z_{k_{max}}$**: Take an example, assume the input is 2D with resolution $100 \times 100$. The input is a $100 \times 100$ matrix. FFT turns it into a $100 \times 100$ frequency  matrix. If we define $k_{max}=10$, then the lower modes $Z_{k_{max}} = $ { $ (k_1, k_2) \mid |k_1|<10, |k_2|<10 $} is the four $10 \times 10$ sub-matrices around the four corners.
> 2. **Parametrization of $R$**: In this case, $R$ is a matrix and its entries are the parameter $\Phi$. We tried other parameterizations to let the entries of R be the output of a linear map or an MLP, as a function of the wavenumber and input function $a(x)$. In these cases, phi is the parameters of the linear map or MLP. In practice, we found the direct parametrization is quite sufficient.
>
> And thanks for catching the above!

---

### Official Review · AnonReviewer2 · 2020-10-30
**Several conceptual questions**

**Rating:** 6
**Confidence:** 2

**Review:**

Paper Summary: The authors proposed a novel neural Fourier operator that generalizes between different function discretization schemes, and achieves superior performance in terms of speed and accuracy compared to learned baselines.

I have to admit that my understanding of this paper is rather limited and I have lots of conceptual questions about the implementation.
- The authors claimed to novel contributions: one is that the method is fast, one is that the method is discretization invariant. These two seem to be in conflict. The authors leveraged FFT for performing fast convolutions differentiably. However FFT assumes the data to be present on a uniform grid (of 2^n grid cells in each dimension). I understand that since the learned content is a frequency "filter" that can be applied to any discretization, but this can be said for just about any CNN (in that case a learned 3x3 filter in physical space can also just be projected to the spectral space and work with "any discretization"). Furthermore (correct me if I'm wrong), it doesn't seem that the authors experimentally illustrated a case of learning on non-discretized data, and performing inference on discretized versions of it, or vice versa.
- From Fig. 1, it seems that the actual parameters that are being learned is the convolutional kernel R (represented as spectral coefficients), and W. That basically amounts to a single convolution plus bias operation. The expressivity of a single such operation should be pretty limited, since this operation itself is a linear operator, while the underlying equations that the authors demonstrated on are mostly nonlinear.

More insights into the two questions above are certainly appreciated.

Adding a reference that seems related to this work. The paper below uses a spectral solver step as a differentiable layer within the neural network for enforcing hard linear constraints in CNNs, also taking the FFT -> spectral operator -> IFFT route.

References
---------------
Kashinath, Karthik, and Philip Marcus. "Enforcing Physical Constraints in CNNs through Differentiable PDE Layer." ICLR 2020 Workshop on Integration of Deep Neural Models and Differential Equations. 2020.

---

> ### Author Response · Authors · 2020-11-23
> **Response to Reviewer 2**
>
> Thank reviewer 2 for the valuable questions:
> 1. **Fast computation and discretization-invariance**: We agree with the reviewer that the FFT is restricted to uniform grids, however, as we described in the paper, the general Fourier transform can be deployed for arbitrary meshes. Since, in our method, we truncate frequency modes up to the $k_{max}$’th ($k_{max}<<n$) frequency modes, both FFT and general Fourier transform methods result in fast computation. Therefore, fast computation and discretization-invariance can be both achieved. Please find the detailed expression in the following.
>     - **Complexity**: The complexity of FFT is $O(n \log n)$ and general Fourier transform is $O(n k_{max})$, where $k_{max}$ is the number of frequency modes. Both are much faster than the full integration ($O(n^2)$ complexity) deployed in the prior works.
>     - **CNN is not mesh-invariant**: Syntax-wise speaking, the CNN can be applied to different resolutions, but when resolution changes the receptive field of the $3 \times 3$ filters changes. Therefore, each resolution requires specific hyper-parameter tuning. As shown in the experiments reported in Fig. 2, the error rate of Fully convolutional networks (FCN) increases with resolution.
>     - **Fourier filters vs CNN filters**: We observe that the Fourier filter is more efficient in representing the solution of PDEs. The filters in CNN are local: they are good to capture local patterns such as edges and shapes from real-world images. On the other hand, the Fourier filters are global: they are better to capture continuous functions in PDEs.
>     - **Non-discretized data**: In experiments, most of the equations don’t have analytic solutions, so we have to use numerical data. We did zero-shot super-resolution in the experiments. For example, we train on $64 \times 64 \times 20$ and test on $256 \times 256 \times 80$, which is presented in Figure 1.
> 2. **Expressivity**: The power of the network comes from combining linear integral operators (via the Fourier transform) and non-linear activation functions. As a result it can learn highly non-linear operators. This is analogous to the standard MLP and CNN where we combine linear multiplications with non-linear activations which allows us to learn highly non-linear functions.
> 3. **Reference**: Thanks for the reference. In the paper "Enforcing Physical Constraints in CNNs through Differentiable PDE Layer." (ICLR 2020 Workshop on PDE), the authors enforce physics constraints within CNNs by projection in the Fourier domain. In our work, we learn the neural operator by doing convolution with Fourier transform. Both involve FFT, but the goals and operations are different. We appreciate the reviewer's suggestion. We have added this reference to the revised paper along with a detailed discussion.
>
> We thank the reviewer for the clarifying comments. We will provide a detailed explanation of these points in the paper.

---

### Official Review · AnonReviewer3 · 2020-11-01
**Review of Fourier Neural Operator for Parametric Partial Differential Equations**

**Rating:** 7
**Confidence:** 3

**Review:**

This work attacks the important problem of learning PDEs with neural networks. To this extent, it proposes a neural network architecture where (in essence), the linearity is not applied in the Fourier space but in the state space. This allows the network to implicitly learn the partial differentials of the equation, and results in a parametrization which is invariant to spatial discretization. This approach yields good results when regressing to data derived from numerical solutions of complex differential equations.


Strengths:

Overall, the paper is well written. The theoretical and experimental sections are, for the most part, clear and concise, although some important details remain unclear/lacking.

The method seems to be quite generic and can be applied to a large range of PDEs.

The inverse problem experiment is interesting and highlights a potentially useful application of the proposed method.


Weaknesses:

Perhaps most importantly, the data generation is not always clear to me. The distinction between training data and test data is not clearly specified. Generalization to different useful initial conditions is obviously of utmost importance in order to properly evaluate the quality of data-driven methods for dynamical systems.

The loss function and training method is not clearly specified: are observations acquired every t=k \in N steps for NS? Do you use teacher forcing or compute an error on the full sequence? What about the other experiments?


Comments and questions:

It seems that in the proposed model, the Fourier coefficients R do  not vary with the layers. This seems to be quite constraining, could you comment on this?

It would have been interesting to analyze the expressiveness of the architecture.

You mention that you are not restricted to periodic boundary conditions due to your final decoder (which is never clearly defined), however, does the finite support of the data not lead to artifacts in the Fourier transform, causing you to have to learn a high number of Fourier modes? Could you comment on this?

In the NS experiment, why have you used N=10 000 when viscosity is set to 1e-4, and N=1000 for the others? It seems as if you have selected the number of samples where your method outperforms the baselines.

Moreover, for the NS experiments, test data seems to be future data (T>10), however this seems to not be the case for the Burgers’ equation, even though it is also time dependent.

As one of your motivations behind this work is to learn the operator, it could have been interesting to test your approach using different sample points as in the training data.

---

> ### Author Response · Authors · 2020-11-23
> **Response to Reviewer 3**
>
> Thanks review 3 for the detailed review and constructive comments! Our response:
>
> 1. **Data generation**: the data generation is given in the Appendix.
>     - **Distribution**: the input initial conditions are sampled from a Gaussian random field. Then we use conventional solvers to obtain ground-truth solutions. Both the training and testing data follow the same distribution. Investigating distributional shift in initial conditions is very important and we plan to pursue it in the future.
>     - **Resolution**: the lower resolution data is downsampled from the higher resolution. In our empirical study, the training resolution and testing resolution are the same, except for the super-resolution where we train on a lower resolution and test on a higher resolution.
> 2. **Training method and loss function**: For NS, the RNN-based methods (Fourier-2d along with other deep learning-based methods), we do the standard training: compute the error for each time-step, and sum them up for later backpropagation. No teacher-forcing. The 3d method (Fourier-3d) directly predicts the full sequence. Other experiments (Darcy & Burgers) are formulated as time-independent problems. For the loss function we use L2 loss.
> 3. **Recurring layers**: The Fourier coefficients R varies with the layers in the models we used in the experiments. However, we empirically observe that having the same R for all the layers, results in faster convergence without much drop in the accuracy.
> 4. **Expressiveness**: Empirically, we observe the Fourier neural operator is more expressive than the baselines, since we observe it achieves the lowest training as well as testing error. In our empirical study we show that the Fourier neural operator approximates highly non-linear solution operators, with non-periodic boundary, and slow spectral decay. We agree with the reviewer that theoretically analyzing the expressiveness of Fourier neural operators is interesting which in fact is our future direction.
> 5. **Periodic boundary condition**: The finite support can cause aliasing, which is alleviated when we truncate the higher frequency modes. But in practice, we do not observe artifacts. Also, we observe the linear transform W outside the Fourier domain helps to capture the non-periodic boundary condition.
> 6. **The number of training samples**: For N=1000, we show that the Fourier-2d method outperforms the baselines on all of 1e-3, 1e-4 and 1e-5. For higher Reynolds numbers, the PDEs are more complicated, and hence more data is required for training. Given a sufficient amount of data, we show that the Fourier-3d method outperforms Fourier-2d.
> 7. **Future data**: In both NS and Burgers, we learn the map from initial condition to later time steps. In Burgers it maps $t=0 \mapsto t=1$; In NS, it maps $t \in [0,10] \mapsto t \in [10,T]$. The settings are the same for both training and testing.
> 8. **Learning operators**: Yes, we do test our approach using different sample points as in the training data. In the experiment on zero-shot super-resolution, where we train on $64 \times 64 \times 20$ and test on $256 \times 256 \times 80$. It’s presented in Figure 1.
>
> We appreciate the reviewer’s detailed comments. We will incorporate them in the paper.

---

### Author Response · Authors · 2020-11-23
**Overall Response**

We want to thank all reviewers for the insightful comments. We are glad that the reviewers agree that machine learning for PDEs is timely and important, and that the results of Fourier neural operators are impressive, leading to many potential applications. We will address the common concerns and then respond to each individual reviewer separately with further details.

1. **Overall impact**: please allow us to restate the impact: Fourier neural operator is the first work that learns the resolution-invariant solution operator for the family of Navier-Stokes equations in the turbulent regime.
    - **Fast**: It is up to three orders of magnitude faster than the conventional pseudo-spectral solver.
    - **Resolution-invariant**: Our method achieves zero-shot super-resolution, meaning it is trained only on low resolution data ($64 \times 64 \times 20$) and evaluated directly in high resolution ($256 \times 256 \times 80$). Previous machine-learning based methods are mostly resolution-dependent, requiring the training and testing to be on the same grid.
    - **Accurate**: Our method still outperforms the baselines by 30% when fixing the resolution in their favor. On the other hand, previous neural operators are resolution-invariant but they did not converge in the challenging case of Navier-Stokes equation.
2. **Architecture diagram**: We add Figure 2 in the revised paper to explain the full architecture. Given the input function, we first lift it to a higher dimension channel space with an encoder network, apply four Fourier layers, then project it back to the desired dimension of the solution function with a decoder network.
3. **Expressiveness**: We empirically show that the proposed architecture can approximate complicated operators arising in PDEs that are highly non-linear, with high frequency modes, and slow spectral decay:
    - **Non-linearity**: The power of the network comes from combining linear integral operators (via the Fourier transform) and non-linear activation functions. As a result, it can learn highly non-linear operators. This is analogous to the standard MLP and CNN where we combine linear multiplications with non-linear activations which allows us to learn highly nonlinear functions.
    - **High frequency modes**: Although we truncate out the higher frequency modes in the Fourier layer, the operator as a whole can approximate functions with the full frequency range. This is because we represent the function in a high dimensional channel space. When projecting back to the desired dimension, the non-linear decoder network recovers the higher frequency modes.
    - **Non-periodic boundary**: the Fourier layer can only approximate functions with periodic boundary, but it is not a limitation of the operator as a whole. The linear transform W outside the Fourier transform (the bias term) keeps track of the non-periodic, location information. Therefore, the full architecture can learn solution functions with non-periodic boundaries.
    - We aim to develop a theoretical analysis for the classes of operators the method can approximate. But approximation of operators is more challenging than that of the functions. We leave this as a future direction.

4. **Revision done to the paper**:
    - Add a paragraph in the introduction to discuss the difference between neural operators and conventional solvers.
    - Add a new diagram to explain the full architecture.
    - Add a discussion of non-periodic boundary conditions.
    - Add details to the experiments: the computation devices, zero-shot super-resolution.
    - Add the references.

---

### Decision · Program_Chairs · 2021-01-07
**Final Decision**

**Decision:**

Accept (Poster)

**Comment:**

Pros:
- Provides a practical technique which can dramatically speed up PDE solving -- this is an important and widely applicable contribution.
- Paper is simultaneously clearly written and mathematically sophisticated.
- The experimental results as impressive.

Cons:
- There were concerns that the paper lacks novelty compared to Li et al 2020b, where the underlying theoretical framework was developed. The primary novelty would seem to be:
- - using Fourier transforms as the specific neural operator
- - the strength of the experimental results

Overall, I recommend acceptance. I believe the techniques in this paper will be practically useful for future research.